# Verification and Evaluation of Male-Produced Pheromone Components from the Citrus Long-Horned Beetle, *Anoplophora chinensis* (Forster) (Insecta: Coleoptera: Cerambycidae)

**DOI:** 10.3390/insects15090692

**Published:** 2024-09-13

**Authors:** Damon Crook, Emily Maynard, Mandy Furtado

**Affiliations:** Forest Pest Methods Laboratory, USDA-APHIS-PPQ, 1398 West Truck Road, Buzzards Bay, MA 02542, USA

**Keywords:** pheromone, *Anoplophora chinensis*, nonanal, 4-(n-heptyloxy)butan-1-ol, 4-(n-heptyloxy) butanal

## Abstract

**Simple Summary:**

The main aim of this study was to identify antennally active pheromone components from the citrus long-horned beetle, *Anoplophora chinensis*, and to test them for behavioral activity. We identified three antennally active male-produced chemical components (nonanal, 4-(n-heptyloxy)butan-1-ol, and 4-(n-heptyloxy)butanal). When tested together in Y-tube behavioral assays, these three compounds were shown to be attractive to adult female *A. chinensis*. Pheromone production and behavior of *A. chinensis* are discussed in relation to other closely related *Anoplophora* cerambycid species.

**Abstract:**

The citrus long-horned beetle, *Anoplophora chinensis* (Coleoptera: Cerambycidae), is a highly polyphagous species native to eastern and southeastern Asia. Detection of these beetles is dependent on visual surveys, which are inefficient and labor-intensive. The identification and development of pheromone-based lures would help improve survey efforts for *A. chinensis* should it become established within the US. We identified three antennally active male-produced chemical components (nonanal, 4-(n-heptyloxy)butan-1-ol, and 4-(n-heptyloxy)butanal), which were then tested for behavioral activity at our USDA Quarantine laboratory. When tested together in Y-tube behavioral assays, a three-component blend of the identified compounds was shown to be attractive to adult female *A. chinensis*. Pheromone production and behavior of *A. chinensis* and other closely related *Anoplophora* cerambycid species are discussed in relation to mate finding and potential monitoring.

## 1. Introduction

The citrus long-horned beetle, *Anoplophora chinensis* (Forster), is a wood-boring cerambycid beetle that is considered a serious pest in its native East Asian range [1]. Unlike many other native borer pests that primarily attack dead trees, *A. chinensis* attacks apparently healthy trees. *A. chinensis* is known to attack over 100 plant species. Its primary hosts include lime/lemon/oranges/tangor (citrus), trifoliate orange (*Poncirus trifoliata*), apple (*Malus pumila*), Australian pine (*Casuarina equisetifolia*), poplars (Populus), and willows (Salix) [2].

It primarily occurs in China, Korea, and Japan, but it is also found in Taiwan, the Philippines, Indonesia, Malaysia, Myanmar, and Vietnam [2]. Fruit trees (particularly Citrus) are weakened by larval feeding, which then become susceptible to wind damage and pathogenic-related diseases. High levels of mortality and economic loss have been reported on young fruit trees as they are particularly susceptible to attack from *A. chinensis* [2].

In North America, there were no records of established populations until infestations were detected for the first time in Tukwila, Washington on maple trees (Acer) imported from Korea [1]. A five-year-long eradication program cost $2.2 million. Prior to the localized Washington population, in 1999 a single adult *A. chinensis* was intercepted at a nursery in Athens, Georgia, on a shipment of crepe myrtle (Lagerstroemia) bonsai from China [3]. An interception of *A. chinensis* in Wisconsin has also occurred [4]. In Europe, *A. chinensis* has proven to be extremely difficult and expensive to eradicate [5]. Populations of *A. chinensis* have become established in Italy (Parabiago 2000; Assago, Milan, and Montichiari 2006; Gussago 2007; and Rome 2008), France (Soyons 2003) and the Netherlands (Het Westland 2007) [5]. Surveying costs alone for *A. chinensis* in Europe between 2001 and 2008 amounted to 2.8 million Euros (3.1 million USD). A closely related species, the Asian long-horned beetle, *Anoplophora glabripennis* (Motschulsky), has caused the destruction of thousands of trees in New York (1996), Illinois (1998), New Jersey (2002), Massachusetts (2008), Ohio (2011), and South Carolina (2020). North American eradication efforts for *A. glabripennis* to date have cost several hundred million dollars [6]. With increasing international trade and movement of plant materials, there is an imminent risk of *A. chinensis* becoming established within the USA. Once established in the USA, *A. chinensis* could become a bigger agricultural problem than *A. glabripennis* due to its broader host range.

To find infested trees and delimit invasive long-horned beetle populations, eradication programs rely on visual surveys, which verify adult presence and damage, such as exit holes and oviposition scars. Heavily infested trees can be easily found, but surveyors are much less efficient at detecting early infestations on trees with minimal damage. Visual inspection of host trees is labor-intensive and relatively inefficient. Improved survey methods could greatly reduce program costs and/or enhance the ability to detect invasive long-horned beetles, especially in areas away from the centers of infestations. Attractant-baited traps provide this function in many insect programs, and prototype traps and lures have been developed for *A. glabripennis* [7,8].

There has been great interest in identifying pheromones of *A. glabripennis* and *A. chinensis* to facilitate their early detection. As of 2012, no long-range pheromone had been reported for either insect, although male-produced short-range pheromones and female-produced contact recognition pheromones had been identified [8,9]. Male *A. glabripennis* produces two functionalized dialkyl ethers, 4-(n-heptyloxy)butanal and 4-(n-heptyloxy)butan-1-ol, which elicit gas chromatography–electroantennogram detection (GC-EAD) responses in females [9] and are attractive to females in laboratory assays [7,9,10]. Moderate attraction in field trapping studies suggested that there may be additional chemical cues that were missing [7,8]. A potential third, male-produced sesquiterpene component was identified for *A. glabripennis* as (3E,6E)-α-farnesene [11]. When (3E,6E)-α-farnesene was combined with 4-(n-heptyloxy)butan-1-ol and 4-(n-heptyloxy)butanal, attraction of both sexes in lab assays increased compared to 4-(n-heptyloxy)butan-1-ol and 4-(n-heptyloxy)butanal alone [11].

Research over the past decade has revealed that pheromone components of cerambycids are often conserved among closely related species (i.e., a chemical can serve as the sole or dominant pheromone component for multiple species). Previous research on *A. chinensis* has suggested that its pheromone is likely to be produced by males, with a reasonable probability that it has the same or a very similar hydroxyether structure to those of its closely related species *A. glabripennis* [12]. Two components of the volatile pheromone produced by males of *A. glabripennis*, 4-(n-heptyloxy)butan-1-ol and 4-(n-heptyloxy)butanal, have been evaluated as potential pheromones of *A. chinensis* [12]. Both compounds were detected in headspace volatiles from male *A. chinensis* but not in volatiles from females. In that study, only 4-(n-heptyloxy)butan-1-ol elicited responses from *A. chinensis* antennae in GC-EAD analyses, and this compound attracted small numbers of adults of both sexes in field trapping studies. These data suggested that 4-(n-heptyloxy) butan-1-ol was an important component of the volatile pheromone produced by males of *A. chinensis*. Volatiles have also been collected and analyzed from male and female adults of a third, closely related Japanese species, *Anoplophora malasiaca* (Thompson), the white-spotted longicorn beetle [13]. This study found 4-(n-heptyloxy)butan-1-ol and 4-(n-heptyloxy)butanal in male extracts as well as nonanal in both male and female volatile extracts. GC-EAD responses showed that nonanal and 4-(n-heptyloxy)butan-1-ol elicited activity from both male and female antennae, but 4-(n-heptyloxy)butanal did not. For *A. malasiaca*, nonanal and 4-(n-heptyloxy)butan-1-ol produced no short-range female attraction in lab behavioral tests. For males, short-range attraction to nonanal was significant at higher doses, but it did not depend on the presence of 4-(n-heptyloxy)butan-1-ol [13]. Based on these recent findings, we wanted to evaluate and verify the presence and antennal activity of nonanal, 4-(n-heptyloxy)butan-1-ol, and 4-(n-heptyloxy) butanal in *A. chinensis* and test any potential behavioral synergy between these three compounds.

## 2. Materials and Methods

### 2.1. Source of Insects

All adult virgin *A. chinensis* (between 3 and 4 weeks old) used in laboratory studies were reared at the Forest Pest Methods Laboratory (FPML) Insect Containment Facility using previously described methods [11]. The insects originated from Italian and Chinese sources. The insects were reared on an artificial diet until adulthood, when they were then fed cut shoots of striped maple (*Acer pensylvanicum*) until needed. Adults were kept in 16 oz plastic Recycleware™ deli cups (Fabri Kal Corporation, Hazle Township, PA, USA). An ice pick was used to puncture two small holes in the deli cup lids. Twigs were placed in the bottom of each cup and replaced every week. All insects were kept at 25 °C, approx. 60% relative humidity, and 16:8 h L:D. The light system above the adult beetles consisted of four T5 fluorescent lamps (Deep Blue Professional, City of Industry, CA, USA). Two were 39 W Solarmax T5 10,000 K daylight lamps, and two were 39 W Solarmax T5 Actinic 03 lamps that emitted a maximum blue phosphor peak at 420 nm. Automatic timers were set so that the actinic lamps turned on at 0630 h and shut off at 2100 h. The daylight lamps turned on at 1030 h and turned off at 1530 h. Light output for the daylight and actinic lamps together was 450 lx.

Individual virgin adults (between 2 and 3 weeks old) were placed in 120 mL glass canning jars (Bell/Uline, Pleasant Prairie, WI, USA) with Teflon screw-on lids that had two openings for tubing. Empty glass jars were aerated and used as controls. Battery-operated pumps (Sensidyne, Clearwater, FL, USA) pulled air through each jar at a rate of 300 mL/min. Ambient air was filtered through a 6–14 mesh activated charcoal (Fisher Scientific, Pittsburgh, PA, USA) inlet before entering the jar. After leaving the jar, the air passed through two traps connected by a short section of Teflon tubing. Each trap was a 3 mm ID × 110 mm glass tube containing 200 mg of 50/80 mesh Super Q (Alltech Associates Inc., Deerfield, IL, USA). All connections were sealed with PTFE sealing tape (Sigma-Aldrich, St. Louis, MO, USA). Aerations were started at 09:00 h and ran for 24 h. A total of 12 male, 12 female, and 12 control aerations were collected. Aeration samples were individually eluted with 2 mL of hexane (HPLC grade, OMNISOLV, Sigma-Aldrich, St. Louis, MO, USA) before being concentrated down to 100 μL under a gentle stream of nitrogen. All extracts were kept at −20 °C before being subjected to GC-MS or GC-EAD analysis.

### 2.2. GC/MS Analyses

Initial chemical analyses were conducted using a combined 6890 network gas chromatograph and 5973 mass-selective detector (Agilent Technologies Inc., Santa Clara, CA, USA). The GC was equipped with a DB-5 column (30 m × 0.25 mm I.D.; film thickness, 0.25 μm; J & W Scientific Inc., Folsom, CA, USA). Helium was the carrier gas at a constant flow rate of 0.7 mL/min. Injection was splitless at 275 °C. The GC oven temperature was held at 40 °C for 1 min, programmed to 270 °C at 10 °C/min, and held for 15 min. Volatiles were identified based on their mass spectra (NIST version 2.0, 2002), Kovats indices [14,15], and comparison of the retention indices and mass spectra with those of available authentic synthetic compounds. Standards (>98% purity) of 4-(n-heptyloxy)butan-1-ol and 4-(n-heptyloxy)butanal (henceforth “alcohol” and “aldehyde”) were obtained from Bedoukian Research Inc. (Danbury, CT, USA). Nonanal (>95% purity) was obtained from Sigma-Aldrich (St. Louis, MO, USA).

### 2.3. Electrophysiological Analysis (GC/EAD)

Samples of aerations or standards (2 μL, 200 ng) were injected in splitless mode onto a Hewlett Packard (Agilent) 6890 gas chromatograph (Agilent Technologies Inc., Santa Clara, CA, USA) with a DB-5MS-DG column (30 m × 0.25 mm ID, 0.25 μm film thickness; J & W Scientific Inc., Folsom, CA, USA) and a 1:1 effluent splitter that allowed simultaneous FID and EAD detection of the separated volatile compounds. Helium was the carrier gas (2.5 mL/min). The GC oven temperature was held at 40 °C for 1 min, programmed to 270 °C at 10 °C/min, and held for 15 min. The injector temperature was 275 °C. The GC outlets for the EAD and FID were maintained at 300 °C. The column outlet for the EAD was held in a humidified air stream flowing at 2 mL/min over the prepared antennae of adult *A. chinensis* attached to an EAG probe (Syntech, Hilversum, The Netherlands). Antennae were prepared by cutting a single antenna at the base of the head of an adult beetle and removing the lower pedicel and scape. An insect pin (size 1) was used to make three holes on the first flagellomere as well as the flagellomere third from the tip. The holes were made deep enough to make a clean opening in the cuticular surface to allow conducting gel (Spectra 360, Parker Laboratories, Fairfield, NJ, USA) to form an uninterrupted connection between the probe and the internal dendrites of the antennae. One of the electrodes on the probe was extended with gold wire (20 mm long) to accommodate the long length of the antennal preparation. This method preserved the tips of the antennae, eliminating the risk of removing vital sensillae specific to that location [12]. The EAG probe was connected to an IDAC-232 serial data acquisition controller (Syntech, Hilversum, The Netherlands). Signals were stored and analyzed on a PC equipped with the EAD program (version 2.6, Syntech).

### 2.4. Olfactometer Assays

A Y-tube olfactometer (Analytical Research Systems Inc., Gainesville, FL, USA) was used to test the biological activity of synthetic samples. All behavioral assays were done in a walk-in environmental chamber (25 °C, approx 60% RH) under a lighting system (4 × T5 Solarmax fluorescent lamps) as described earlier. The Y-tube was held at a 15° angle upward from horizontal on a custom-built holder placed 0.5 m below the lighting (measured at 300 lx). A custom-cut Y-tube was used, constructed from a Teflon block measuring 32 cm tall, 30 cm wide, and 4.5 cm deep. The main arm of the Y measured 13 cm, and the two top arms were 15.5 cm long and 5.1 cm wide. The top arms were angled at 90°. The openings in the two top arms for tubing were 1.8 cm in diameter. The opening in the bottom of the main arm was 3.2 cm in diameter. The point of entrance of the two top arms was 16 cm from the bottom of the Y junction. Each arm was then connected to a separate Nalgene tube containing either the stimulus or a solvent/blank control. Charcoal-filtered air was bubbled through distilled water and then into each of the two arms at 1.0 L/min using a 2-channel air delivery system (Analytical Research Systems Inc., Gainesville, FL, USA). Male and female adults between 14 and 97 days old were used for all olfactometer bioassays. Individual beetles used in these bioassays were from the lab colony. Insects were feeding on twigs until used in tests (i.e., no starvation period). A total of 20 replicates were completed for each treatment, using one beetle per replicate. Stock solutions of the alcohol, aldehyde, and nonanal (10 μg in 10 μL hexane) were used for all tests. Bioassays were conducted to test the attraction of the combination (1:1:1) of alcohol, aldehyde, and nonanal at 0.1 μg, 1 μg, and 10 μg doses. The three components were also tested individually at 1 μg dose. Doses were selected based on previous olfactometer bioassays involving *A. glabripennis* [7,9]. Treatments were offered against hexane as the control. All combinations and concentrations of the test stimulus were conducted on both male and female beetles. The test stimulus was dispensed onto a strip of filter paper (10 × 40 mm) and placed in the tube connected to one arm of the olfactometer. An identical filter paper strip with the same amount of hexane was placed in the other arm of the olfactometer to act as the control. The Y-tube was rinsed with acetone between each individual test. Treatment and control arms were alternated every other replicate to control for possible positional effects. For each test, a single male or female beetle was placed at the end of the main stem and given 5 min to choose between the two stimuli. Insects were considered to have made a choice when their entire body crossed the point of entrance at either top arm. No choice was recorded if the beetle failed to pass either line after the 5 min period. Insects that did not make a choice were excluded from the statistical analysis. All experiments were conducted between 1100 h and 1500 h, when beetles appeared to be most active.

### 2.5. Statistical Analyses

To test whether the test stimulus attracted more beetles than the solvent control in Y-tube olfactometer bioassays, a X^2^ analysis goodness-of-fit test was used. Values of X^2^ > 3.84 with 1 d.f were considered significant (*p* < 0.05).

## 3. Results

### 3.1. Analysis of Aeration Extracts

Aerations of male extracts yielded 4-(n-heptyloxy)butan-1-ol, 4-(n-heptyloxy)butanal, and nonanal. Aeration extracts from females produced nonanal but did not yield either the alcohol or the aldehyde (Figure 1). All three compounds elicited antennal responses from both females and males. When standards of the three products were used, antennal responses from both male and female beetles were more distinct (e.g., Figure 2).

### 3.2. Olfactometer Bioassays

In Y-tube olfactometer bioassays, female beetles were significantly more attracted to the three-component blend of the alcohol, the aldehyde, and nonanal compared to the control (Table 1). Neither any two-component blend nor the alcohol, aldehyde, or nonanal alone was significantly more attractive compared to the control (Table 1). The female bioassays did show a dose-dependent trend, with peak attraction occurring at the 1 ug dose (Table 2). Males were not significantly attracted to any blend over the control (Table 1). Dose had no effect on male attraction to the three-component blend (Table 2).

## 4. Discussion

In an effort to better understand male-produced attractants, particularly those already identified in *Anoplophora* spp., we investigated the headspace volatiles of both male and female *A. chinensis*. We identified 4-(n-heptyloxy)butan-1-ol and 4-(n-hepty)butanal in male aeration products. We also identified nonanal as a major component in both male and female headspace volatiles of *A. chinensis*. Our GC-EAD results show that the alcohol, the aldehyde, and nonanal all elicited antennal responses from male and female beetles. Our results differ from a previous study on *A. chinensis* [12], which found that the aldehyde component did not elicit an antennal response from either sex. These differing GC-EAD results are probably attributed to slight methodological differences in the way antennae were prepared in each study.

Research on *A. malasiaca* also found the same alcohol and aldehyde components to be produced by male adults [13]. They also detected nonanal in male and female volatile extracts. Their GC-EAD analysis demonstrated antennal responses to nonanal and the alcohol component by both male and female *A. malasiaca*. No antennal responses were observed for 4-(n-heptyloxy)butanal by either sex in that study. In our lab behavioral assays with *A. chinensis*, we found that females are not attracted to 4-(n-heptyloxy)butan-1-ol or 4-(n-heptyloxy)butanal when tested individually or as a two-component blend. Males were not attracted to either 4-(n-hepty)butanal alone or a two-component blend. Male beetles did display a preference for 4-(n-heptyloxy)butan-1-ol in Y tube assays (70%), but their attraction was not statistically significant (*p* = 0.07364). This result may indicate the male-produced 4-(n-heptyloxy)butan-1-ol is potentially used as an aggregation pheromone by *A. chinensis*, a conclusion suggested after testing lures in field traps [12]. Field traps baited with 4-(n-heptyloxy)butan-1-ol alone or in a 1:1 blend with 4-(n-heptyloxy)butanal caught significantly more beetles of both sexes than those baited with 4-(n-heptyloxy)butanal alone or control traps [12].

In terms of behavioral responses of *A. malasiaca* to 4-(n-heptyloxy)butan-1-ol, 4-(n-heptyloxy)butanal, and nonanal, males showed a dose-dependent response to nonanal at dosages over 10 ng [13]. Female attraction to 4-(n-heptyloxy)butan-1-ol was found to be dose-dependent; however, female and male responses to the alcohol were not statistically different from that of control. In our Y-tube assays, we found neither sex of *A. chinensis* was attracted to nonanal or any two-component blend with nonanal. Females were only attracted to the three-component blend of 4-(n-heptyloxy)butan-1-ol, 4-(n-heptyloxy)butanal, and nonanal.

Both 4-(n-heptyloxy)butan-1-ol and 4-(n-heptyloxy)butanal have also been identified in *A. glabripennis*, further suggesting male-produced pheromones are conserved within the genus [9,16]. There is also some evidence to suggest that “less volatile” contact pheromones may also be shared within the genus. Mating behavior studies between *A. glabripennis* and *A. malasiaca* have suggested that there are contact cues on the elytra of *A. glabripennis* that can elicit mating attempts by male *A. malasiaca* [17].

Host volatiles may play an important synergistic role in the attraction of *A. chinensis* to male-produced pheromones (4-(n-heptyloxy)butan-1-ol and 4-(n-heptyloxy)butanal). It has been repetitively found that *Apoplophora* spp. female catch increases when host volatiles are added, while males show a preference for host volatiles alone [16,18,19,20]. A recent study in Italy [20] tested commercially available lures designed for *A. glabripennis* (lures comprised the two male pheromone components and host volatiles, which included (-)-linalool and (Z)-3-hexen-1-ol). All three lures performed similarly, with the lure containing beta-caryophyllene catching the most *A. chinensis* adults.

Another study [7] showed that the attraction of male *A. glabripennis* adults was doubled when (-)-linalool or (-)-linalool and (Z)-3-hexen-1-ol were combined with 4-(n-heptyloxy)butan-1-ol and 4-(n-heptyloxy)butanal. Another field trapping of *A. glabripennis* showed that significantly more females were caught in traps baited with a mixture of (-)-linalool, cis-3-hexen-1-ol, linalool oxide, trans-caryophyllene, and trans-pinocarveol, along with the male pheromone components [18]. In additional field trapping [19], more female *A. glabripennis* were caught when (-)-linalool, trans-caryophyllene, and (Z)-3-hexen-1-ol were added to the male pheromone components compared to pheromone or host volatiles alone. In a field experiment on *A. chinensis,* the authors found that host volatiles added to the male pheromone caught adults at the same rate, with 4-(n-heptyloxy)butan-1-ol + 4-(n-heptyloxy)butanal + camphene + cis-3-hexen-1-ol + ocimene + β-caryophyllene and *Melia azedarach* volatiles performing the best [16]. This study examined the volatiles of three host plants (*Acer negundo*, *Salix babylonica*, and *Melia azedarach*). The authors reported that *A. glabripennis* and *A. chinensis* both responded to several common host volatiles identified in the three host plants. The plant volatiles were identified as nonanal, camphene, cis-3-hexen-1-ol, ocimene, and β-caryophyllene. They also found that *A. glabripennis* and *A. chinensis* both responded to 4-(n-heptyloxy)butan-1-ol and 4-(n-heptyloxy)butanal. Although that study identified nonanal as a host volatile, our study detected nonanal as a volatile produced by the insect itself. Nonanal was also previously reported as a female product in *A. glabripennis* [8]. We, therefore, conclude that nonanal is potentially an important component in the general chemical ecology of *Anoplophora* species. Further field test studies are needed to determine if nonanal can improve current detection methods, which utilize pheromone and host volatile-based lures.

## Figures and Tables

**Figure 1 insects-15-00692-f001:**
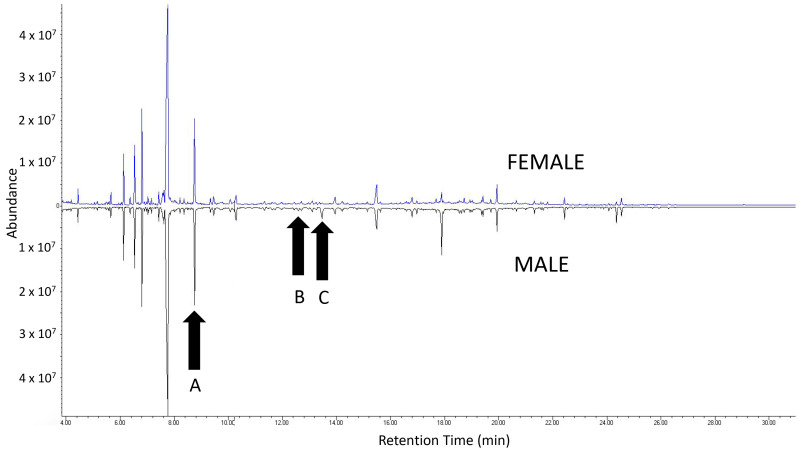
Gas chromatography–mass spectrometry analysis and comparison of aeration samples from female and male adult *A. chinensis*. A = nonanal; B = 4-(n-heptyloxy)butanal; C = 4-(n-heptyloxy)butan-1-ol.

**Figure 2 insects-15-00692-f002:**
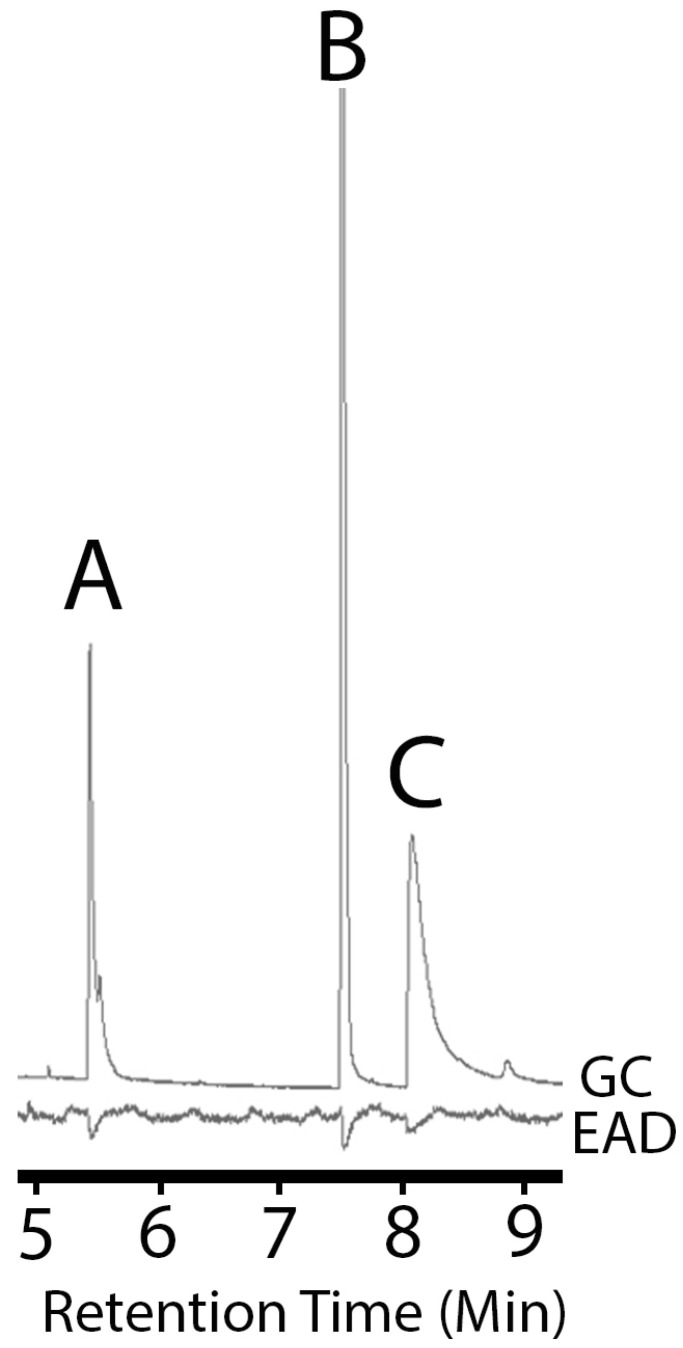
Coupled gas chromatography–electroantennogram detection by *A. chinensis* (female) in response to chemical standards of identified potential pheromone components. A = nonanal; B = 4-(n-heptyloxy)butanal; C = 4-(n-heptyloxy)butan-1-ol.

**Table 1 insects-15-00692-t001:** Response of *Anoplophora chinensis* to male volatiles in a Y-tube olfactometer.

Sex of Beetle	Treatment Source	No. Chosen Treatment	Control Source	No. Chosen Control	% Response to Treatment	X^2^	*p*
Female	Alcohol, Aldehyde, Nonanal	16	Hexane	4	80	7.2	<0.05
Female	Alcohol, Aldehyde	7	Hexane	13	35	1.8	0.1797
Female	Alcohol, Nonanal	10	Hexane	10	50	0	1
Female	Aldehyde, Nonanal	10	Hexane	10	50	0	1
Female	Alcohol	8	Hexane	12	40	0.8	0.3711
Female	Aldehyde	10	Hexane	10	50	0	1
Female	Nonanal	13	Hexane	7	65	1.8	0.1797
Male	Alcohol, Aldehyde, Nonanal	11	Hexane	9	55	0.2	0.6547
Male	Alcohol, Aldehyde	9	Hexane	11	45	0.2	0.6547
Male	Alcohol, Nonanal	10	Hexane	10	50	0	1
Male	Aldehyde, Nonanal	12	Hexane	8	60	0.8	0.3711
Male	Alcohol	14	Hexane	6	70	3.2	0.07364
Male	Aldehyde	8	Hexane	12	40	0.8	0.3711
Male	Nonanal	12	Hexane	8	60	0.8	0.3711

**Table 2 insects-15-00692-t002:** Dose response of *Anoplophora chinensis* to male volatiles in a Y-tube olfactometer.

Sex of Beetle	Treatment Source	Dose (μg)	No. Chosen Treatment	Control Source	No. Chosen Control	% Response to Treatment	X^2^	*p*
Female	Alcohol, Aldehyde, Nonanal	0.1	11	Hexane	9	55	0.2	0.6547
Female	Alcohol, Aldehyde, Nonanal	1	16	Hexane	4	80	7.2	<0.05
Female	Alcohol, Aldehyde, Nonanal	10	10	Hexane	10	50	0	1
Male	Alcohol, Aldehyde, Nonanal	0.1	11	Hexane	9	55	0.2	0.6547
Male	Alcohol, Aldehyde, Nonanal	1	11	Hexane	9	55	0.2	0.6547
Male	Alcohol, Aldehyde, Nonanal	10	7	Hexane	13	35	1.8	0.1797

## Data Availability

Data are available upon request.

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
