# Peer review of "Verification and Evaluation of Male-Produced Pheromone Components from the Citrus Long-Horned Beetle, Anoplophora chinensis (Forster) (Insecta: Coleoptera: Cerambycidae)"

_insects, 2024, doi:10.3390/insects15090692_

Round 1
Reviewer 1 Report
Comments and Suggestions for Authors
1. It is recommended to revise the title, as it is not meant for identification purposes but rather for evaluating the previously identified compounds as potential pheromones.
2. While the researchers possess GC-EAD equipment, the results of the electroantennographic response of the aeration were not presented; only the synthetic compounds were assessed.
3. For Figure 1, add time units to the x-axis and transform the y-axis to represent the percentage of abundance.
4. In Table 2, replace 'x2' with 'chi square'.
5. Ensure to discuss the findings of Yasui, 2023 (https://doi.org/10.3390/insects14020171) in the discussion section.
Author Response
Reviewer 1
Comments and Suggestions for Authors
- It is recommended to revise the title, as it is not meant for identification purposes but rather for evaluating the previously identified compounds as potential pheromones.
Response- I agree with this great suggestion and have changed the title to “Verification and Evaluation….”
- While the researchers possess GC-EAD equipment, the results of the electroantennographic response of the aeration were not presented; only the synthetic compounds were assessed.
Response- Responses to the very low concentrations found in aerations were not as distinct as standards at a higher dosage. We do state that clearer, more obvious responses were seen for the standards. As these compounds have already been identified as potential pheromones we did not think this would be an issue. We just wanted clear responses as a previous publication did not get responses for all three compounds.
- For Figure 1, add time units to the x-axis and transform the y-axis to represent the percentage of abundance.
Response -Fig 1 has a “Retention time” axis label and the Y axis has the standard GC axis “Abundance”.
- In Table 2, replace 'x2' with 'chi square'.
Response – I have changed X2 to chi-square
- Ensure to discuss the findings of Yasui, 2023 (https://doi.org/10.3390/insects14020171) in the discussion section.
Response- I have added in the citation as it relates to contact pheromones possibly being shared between closely related species.
Reviewer 2 Report
Comments and Suggestions for Authors
Straightforward, can't really add much.
A field test would have been nice, but would have required a collaborational effort with a foreign group. Meanwhile, an olfactometer assay is a first, acceptable approach.
Chemical purity of test compounds can be added to M&M.
Table 2. Uncessary to repeat "aldehyde…" in each line.
Line 104. delete "based"
284. the ones
Author Response
Reviewer 2
Comments and Suggestions for Authors
Straightforward, can't really add much.
A field test would have been nice, but would have required a collaborational effort with a foreign group. Meanwhile, an olfactometer assay is a first, acceptable approach.
Chemical purity of test compounds can be added to M&M.
Response- Ive added % Purity to M and M.
Table 2. Uncessary to repeat "aldehyde…" in each line.
Response – The Table is a bit repetitive but why delete just aldehyde? Should we change it to ALC/ALD/NON ? I am open to suggestions from Editor on how to best present that.
Line 104. delete "based"
Response – I agree. I’ve deleted “based”.
- the ones
Response- not sure whats needed here???
Reviewer 3 Report
Comments and Suggestions for Authors
This study aimed to evaluate the compounds released by male Anoplophora chinensis, which are known to elicit behavioral and physiological responses in conspecifics. The findings conclude that a blend of three specific compounds effectively triggers behavioral responses in females.
General comments
The manuscript requires significant improvement in its writing structure and the transitions between phrases throughout. As currently written, it gives the impression that the study was not thoroughly planned, despite the authors employing appropriate methods to evaluate the air-entrapped volatiles. The introduction is overly long. Including a paragraph on kairomone responses in cerambycid beetles is essential, as it would support the response of Nonanal in a three-compound blend. Nonanal and other host plant volatiles are well-studied in cerambycids, with documented evidence of both behavioral and electroantennogram (EAG) responses. The list of compounds evaluated does not include others known to elicit responses in closely related species. To substantiate the claim that the identified compounds function as pheromones, I recommend that the authors evaluate compounds from closely related species which would serve as a standard check and generate EAG dose-response curves for all three compounds and the blend.
Specific suggestions
L17-22: “with the increase…..closely related Asian longhorned beetle, Anoplophora glabripennis”. This section can be moved to the introduction. Please ensure that the study's key findings are emphasized in the abstract.
L48-67: It can be summarized in a few sentences without losing its core essence.
L156-157: Elaborate on the dilutions.
L178: IDAC 2 or IDAC 32
L200-203: I find it puzzling that the authors did not consider replicating the GC peak areas as ratios, rather than defaulting to a 1:1:1 ratio. Additionally, comparing the blend to the GC peak ratios would have been essential to determine whether the female response was specifically ratio-dependent or if the compound ratios were non-specific
L219-221: Suggest an GLM with binomial distribution
L230: Arrow B points to a peak that is unclear and requires revision.
L234: A. chinensis or A. malasiaca?
L246: Revise the table caption to indicate that the data reflects a dose-dependent response.
Comments on the Quality of English LanguageThe manuscript requires significant improvement in its writing structure and the transitions between phrases throughout.
Author Response
Reviewer 3
Comments and Suggestions for Authors
This study aimed to evaluate the compounds released by male Anoplophora chinensis, which are known to elicit behavioral and physiological responses in conspecifics. The findings conclude that a blend of three specific compounds effectively triggers behavioral responses in females.
General comments
The manuscript requires significant improvement in its writing structure and the transitions between phrases throughout. As currently written, it gives the impression that the study was not thoroughly planned, despite the authors employing appropriate methods to evaluate the air-entrapped volatiles. The introduction is overly long. Including a paragraph on kairomone responses in cerambycid beetles is essential, as it would support the response of Nonanal in a three-compound blend. Nonanal and other host plant volatiles are well-studied in cerambycids, with documented evidence of both behavioral and electroantennogram (EAG) responses. The list of compounds evaluated does not include others known to elicit responses in closely related species. To substantiate the claim that the identified compounds function as pheromones, I recommend that the authors evaluate compounds from closely related species which would serve as a standard check and generate EAG dose-response curves for all three compounds and the blend.
Response- With all due respect to Reviewer 3 I think they do not fully appreciate the main aims of this paper.
The main aim was to verify previous studies on this beetle (and closely related species) with regards to what males (and females) were producing as active volatiles. Nonanal is indeed a common host volatile but our insect aerations show that the insect itself is emitting it in some way. Our goal was to verify the presence of these three previously identified insect produced compounds and see how they behaved to them under lab conditions. We hoped that this will clarify (to a degree) how three different species utilize the same ‘pheromonal’ components. The aim was not to test a broad range of host volatiles or other cerambycid pheromones to magically develop some super lure blend. If other cerambycid ‘known pheromones’ are not appearing in our aerations why would I run EAD on standards of them for this species. It could be interesting to see if other closely related pheromones stimulate CLB antennae but that is work for another project. The same applies to screening multiple host volatiles…which researchers have had very little luck with for over 30 years.
The last 30 lines of the paper discusses Anoplophora with regards to host volatiles in depth and nonanal is reported there. We have clearly stated that nonanal is merely an important component of the chemical ecology of this species group and left it at that. More work is needed….
Because the Discussion covers host volatiles in detail (especially with regards to nonanal) we decided not to increase the size of the Introduction any further. The main aim is insect volatile clarification so we feel the overall structure is better if left alone. The other three reviewers liked the writing and had very little to add so I am a little puzzled by this one reviewers comments.
Specific suggestions
L17-22: “with the increase…..closely related Asian longhorned beetle, Anoplophora glabripennis”. This section can be moved to the introduction. Please ensure that the study's key findings are emphasized in the abstract.
Response- Thankyou for this suggestion. We agree that it is not needed in the abstract and distracts from main points.
L48-67: It can be summarized in a few sentences without losing its core essence.
Response- I’ve edited down the detection history section a little.
L156-157: Elaborate on the dilutions.
Response added 200ng per standard 2ul.
L178: IDAC 2 or IDAC 32
Response – my unit says 232.
L200-203: I find it puzzling that the authors did not consider replicating the GC peak areas as ratios, rather than defaulting to a 1:1:1 ratio. Additionally, comparing the blend to the GC peak ratios would have been essential to determine whether the female response was specifically ratio-dependent or if the compound ratios were non-specific
Response- If we were developing release rates for field lures we would have started to investigate ratios in more detail. Ratios differed slightly between aeration samples so we stuck with a 1:1:1 ratio just to show evidence of a behavioral interaction for females. We could certainly examine ratios further if we do want to test field lures or wanted to try and improve attraction somehow.
L219-221: Suggest an GLM with binomial distribution
Response Not really necessary for a two choice assay and we have published simple Y tube stats before with no issues. Other reviewers were also ok with analysis as well as our USDA stats person. Model seemed like overkill for same results.
L230: Arrow B points to a peak that is unclear and requires revision.
Response- Can we increase size to make it clearer? It is small but obviously male only compared to female GC trace.
L234: A. chinensis or A. malasiaca?
Response – Thank you for noticing
L246: Revise the table caption to indicate that the data reflects a dose-dependent response.
Response – Done!
Reviewer 4 Report
Comments and Suggestions for Authors
This manuscript describes behavioural and physiological studies to attempt to further elucidate the pheromones of Anoplophora chinensis. Thank you for the opportunity to review this very interesting manuscript, I trust that it will increase our understanding of the chemical ecology of this important cerambycid genus, which is an actual or potential pest around the world. The manuscript is well-written and easy to read, uses appropriate laboratory-based behavioral and electrophysiological assays, and the data are analysed in an appropriate manner. I hope that, unlike some previous studies, the laboratory trials relate well to field studies, which have to date had disappointing results for this genus.
I have just a few minor points that should be addressed before publication (see below)
Ln 67 replace “…due it’s larger…” with “…due to its larger…”, noting both the addition of the word "to" and the correction of the spelling of "its"
Ln 228 Figure 2 only shows the response of a female, rather than both sexes, I recommend changing “both male and female beetles were more distinct (Figure 2)” to “both male and female beetles were more distinct (e.g. Figure 2)”
Ln 234 I assume that you have inadvertently replaced the name A. chinensis with A. malaisica, but please use the correct species name
Ln 267 this may be due to the limited sample size resulting in insufficient power in the test, increasing the number of beetles tested may show that the results with males and the alcohol is valid, but not detected here due to this
Ln 289 remove the word “were” it adds nothing here and is confusing
Author Response
Reviewer 4
Comments and Suggestions for Authors
This manuscript describes behavioural and physiological studies to attempt to further elucidate the pheromones of Anoplophora chinensis. Thank you for the opportunity to review this very interesting manuscript, I trust that it will increase our understanding of the chemical ecology of this important cerambycid genus, which is an actual or potential pest around the world. The manuscript is well-written and easy to read, uses appropriate laboratory-based behavioral and electrophysiological assays, and the data are analysed in an appropriate manner. I hope that, unlike some previous studies, the laboratory trials relate well to field studies, which have to date had disappointing results for this genus.
Response Thank you for your review and kind comments.
I have just a few minor points that should be addressed before publication (see below)
Ln 67 replace “…due it’s larger…” with “…due to its larger…”, noting both the addition of the word "to" and the correction of the spelling of "its"
Response. Well spotted. Thankyou!
Ln 228 Figure 2 only shows the response of a female, rather than both sexes, I recommend changing “both male and female beetles were more distinct (Figure 2)” to “both male and female beetles were more distinct (e.g. Figure 2)”
Response- Thankyou for suggestion. I have changed line 230.
Ln 234 I assume that you have inadvertently replaced the name A. chinensis with A. malaisica, but please use the correct species name
Response- Thankyou for noticing that error!
Ln 267 this may be due to the limited sample size resulting in insufficient power in the test, increasing the number of beetles tested may show that the results with males and the alcohol is valid, but not detected here due to this.
Response- I would agree with that comment. We do point out the 70% attraction and potential of it on line 280-284.
Ln 289 remove the word “were” it adds nothing here and is confusing
Response – edited accordingly.
Round 2
Reviewer 1 Report
Comments and Suggestions for Authors
In Figure 1, enhance the resolution (retention time is indistinguishable). Convert the abundance into a percentage.
In Figure 2, address the same issues highlighted in the preceding point. Additionally, include the EAD response scale (mV).
In Table 2, utilize the ChiSquare symbol instead of the word.
Author Response
In Figure 1, enhance the resolution (retention time is indistinguishable). Convert the abundance into a percentage.
Figure has been corrected in terms of resolution. I've uploaded it as a separate file. We have not converted to % Abundance as we are comparing overall abundance between two chromatograms (male and female).
In Figure 2, address the same issues highlighted in the preceding point. Additionally, include the EAD response scale (mV).
Figure 2 has also been corrected in terms of better resolution. mV scale is not available since we switched software and a hard drive since this work was done. For simple YES/NO responses on ID publications it usually isnt needed (unless you do dose response curves).
In Table 2, utilize the ChiSquare symbol instead of the word.
Response done
Reviewer 3 Report
Comments and Suggestions for Authors
Thank you for revising the manuscript. A few suggestions to enhance the quality of the manuscript.
Comments on the Quality of English Language
The language used in the manuscript could benefit from some improvement. The transitions between phrases and sentences are somewhat loosely formed, which can affect the overall clarity and flow. Enhancing these transitions would help to better convey the study’s findings and arguments.
Author Response
Reviewer comment.
The language used in the manuscript could benefit from some improvement. The transitions between phrases and sentences are somewhat loosely formed, which can affect the overall clarity and flow. Enhancing these transitions would help to better convey the study’s findings and arguments.
Response- I have carefully read through the latest draft and feel its order and structure is fine. The Introduction covers several topics in an ordered structure......1. Pest status and distribution/risk. 2. Issues with surveys. 3. Brief History of lure development. 4. Conservation of similar attractants between closely related beetles and main goal of this paper.
Methods - follow the structure and form of many other similar chemical ecology manuscripts.
Discussion- Has a summary of our results and compares them with recent research on similar species. We end with possible synergistic effects of host volatiles with respect to these beetles and our latest findings. Its not going back and forth over separate topics.
The three other reviewers rated the writing quality as very good. Two required no changes to English and one had some minor changes to typos. I have ran this draft by two chemical ecologists with over 70 years experience publishing experience. They both feel that the reviewer's comments are a little unfair, especially as the reviewer does not give any examples of 'loosely structured sentences" to justify his opinion.
I will not be editing this draft further without precise reviewer examples that the Editor agrees with.